# REASONING AT THE RIGHT LENGTH: ADAPTIVE BUDGET FORCING FOR EFFICIENT AND ACCURATE LLM INFERENCE

## ABSTRACT

Large Language Models (LLMs) face persistent challenges in domain-specific reasoning tasks, particularly in fields such as mathematics, telecommunications, and scientific problem-solving, where structured, multi-step inference and adherence to formal constraints are essential. Traditional generation strategies often fail to balance accuracy and efficiency in these settings, partly due to the absence of high-quality, curated reasoning datasets. In this work, we introduce **Adaptive Budget Forcing (ABF)**, a simple but very effective test-time inference strategy that dynamically adjusts the reasoning length of LLMs by monitoring real-time certainty signals—such as token-level confidence, entropy, and semantic coherence—within the model's thinking trajectory. ABF enables models to terminate generation when sufficient confidence is reached or extend it when further inference is needed, improving both computational efficiency and decision fidelity. To support ABF and enable effective fine-tuning, we construct **TCORE (Telecom-Curated Open Reasoning Examples)**, a domain-specific dataset featuring multi-step reasoning traces derived from telecom standards and engineering tasks. TCORE is built via a multi-stage filtering process targeting quality, difficulty, and semantic diversity, and serves as both a fine-tuning resource and evaluation benchmark. Experimental results on telecom and mathematical reasoning tasks demonstrate that ABF consistently improves reasoning accuracy while reducing unnecessary computation. TCORE and code are available at `https://anonymous.4open.science/r/ABF-TCORE`.

## 1 INTRODUCTION

Large Language Models (LLMs) have achieved remarkable success across a wide range of natural language tasks, from open-domain question answering to code generation and translation (Brown et al., 2020; Touvron et al., 2023). However, their effectiveness diminishes in structured, domain-specific reasoning settings such as mathematics, scientific problem-solving, and telecommunications. These domains require multi-step logical inference, numerical precision, and adherence to formal specifications (Wei et al., 2022; Cobbe et al., 2021; Li et al., 2023)—capabilities that general-purpose LLMs often lack.

A key limitation in such settings is inference inefficiency. LLMs frequently display reasoning pathologies: overthinking—producing excessively long, redundant chains—or underthinking—terminating prematurely without sufficient justification (Yao et al., 2023a; Muennighoff et al., 2025). Both degrade accuracy and inflate computational costs, primarily when correctness depends on intermediate reasoning. These issues are exacerbated by miscalibrated uncertainty estimates and the absence of principled halting rules (Lightman et al., 2023; Wu et al., 2024). Recent proposals such as budget forcing (Muennighoff et al., 2025) and reward-guided reflection (Wu et al., 2024) attempt to regulate reasoning dynamically but apply fixed heuristics or require additional training overhead.

We propose **Adaptive Budget Forcing (ABF)**, a lightweight yet effective test-time inference strategy that formulates reasoning termination as an optimal stopping problem. As shown in Fig 1, at each generation step, ABF computes a composite certainty score that combines token-level con-

Figure 1: Traditional BF applies a static token budget, while ABF monitors a composite certainty score $C_t$ at each step and adaptively decides whether to continue ([Wait]) or terminate ([Final Answer]).

fidence, entropy, and semantic coherence, and halts when evidence suggests further reasoning is unlikely to change the outcome. This rule-based policy is motivated by decision-theoretic stopping criteria and avoids both redundant computation and premature truncation. Unlike static budgets, ABF adapts reasoning effort to problem difficulty while incurring minimal overhead.

To demonstrate ABF in practice, we conduct extensive experiments across different benchmarks. We also release **TCORE (Telecom-Curated Open Reasoning Examples)**, a carefully filtered dataset of multi-step telecom reasoning traces. While not intended as a new large-scale benchmark, TCORE serves as a proof-of-concept case study illustrating how domain-specific supervision can complement ABF. We provide ablation studies, runtime analysis, and comparisons against learned halting baselines to substantiate the effectiveness of our approach.

Our contributions are as follows:

- We introduce ABF, a general-purpose, decision-theoretic halting strategy that dynamically allocates reasoning length at inference time.
- We provide a theoretical motivation for ABF's composite certainty score and empirically validate its components through detailed ablations.
- We evaluate ABF across multiple benchmarks, showing consistent accuracy gains and 20-30% token savings with negligible runtime overhead.
- We release TCORE, a curated telecom reasoning corpus that serves as a domain-specific case study for ABF, along with code for full reproducibility.

The rest of the paper is organized as follows. Section 2 reviews test-time scaling and adaptive reasoning. Section 3 introduces ABF in detail with theoretical grounding. Section 4 presents the construction of TCORE. Section 5 reports results, ablations, and runtime analysis. Section 6 concludes with limitations and future directions.

## 2 RELATED WORK

**Adaptive Reasoning Strategies.** Several approaches refine LLM reasoning by structuring the thinking process and selectively expanding depth when needed. Teng et al. (2025) decompose complex problems into atomic subquestions, while Li et al. (2025) align reasoning-as-logic-units to reduce hallucinations through program–language correspondence. Beyond explicit decomposition, recent work learns how much to think: Meta Chain-of-Thought introduces meta-level control of intermediate reflections (Xiang et al., 2025), Inner Thinking Transformer scales internal depth adaptively (Chen et al., 2025), and Entropy Adaptive Decoding switches models on the fly based on uncertainty (Simonds, 2025). Hesitation-aware reframing can also recalibrate inference when the model "pauses" (Storaï & Hwang, 2024), and certainty-driven controllers modulate budget by predicted confidence (Nogueira et al., 2025). Outside purely internal control, acting frameworks such as ReAct integrate tool use with stepwise reasoning (Yao et al., 2023b), while retrieval augments reasoning with external knowledge (Lewis et al., 2020; Borgeaud et al., 2022; Shi et al., 2023). These

directions collectively point to structured and adaptive mechanisms that govern which subproblems to solve, how deeply to deliberate, and when to seek external information.

**Inference Efficiency and Budget-Aware Evaluation.** A parallel literature makes computation an explicit first-class objective. Wang et al. (2024) propose budget-aware evaluation spanning tokens and monetary cost, arguing that token-normalized metrics better reflect practical efficiency. Building on this lens, Han et al. (2024) introduces token-budget-aware policies that adapt generation length to balance cost and accuracy. Test-time scaling methods show that controlling thinking depth can rival scaling parameters: s1 demonstrates simple budget forcing and stop-suppression yield significant math gains (Muennighoff et al., 2025), and principled analyses/theory indicate that optimally allocating inference compute can outperform parameter scaling (Snell et al., 2024; Wu et al., 2024). Recent systems further operationalize planning and allocation—e.g., planning-and-budgeting controllers (Lin et al., 2025a), budget-aware control tokens (Wen et al., 2025), and evolving meta-thoughts (Liu et al., 2025). In production settings, dynamic depth and pruning schemes provide complementary efficiency levers at the architectural level (Lin et al., 2025b). Together, these works motivate budget awareness as a core design and evaluation principle: report accuracy and compute, and design policies that spend tokens where they matter most.

**Reflection and Deliberation in LLMs.** A third line of work explicitly injects iterative self-critique, verification, and search. Tree-of-Thoughts organizes multi-branch exploration with intermediate checkpoints (Yao et al., 2023a); process-level verification and step-by-step checking improve logical fidelity and reduce compounding errors (Lightman et al., 2023). REBASE learns a process reward over intermediate steps to steer deliberation through structured reasoning trees (Wu et al., 2024). Complementary evidence shows that simple, lightweight test-time interventions can reliably extend and improve chains of thought: budget forcing in s1 increases solution rates on competitive math without full retraining (Muennighoff et al., 2025); Quiet-STAR illustrates that models can self-induce better pre-output thinking via self-training (Zelikman et al., 2024). More broadly, system-level studies argue for jointly advancing RL-based post-training and inference scaling to strengthen reasoning under tight budgets (Hou et al., 2025).

# 3 METHODOLOGY

## 3.1 BUDGET FORCING

LLMs often waste inference compute on domain tasks with heterogeneous difficulty: some inputs need only a few steps, while others require deeper exploration. Fixed decoding limits or uniform sampling counts lead to needless tokens (overthinking) or premature truncation (underthinking), harming accuracy and efficiency.

**Issue with Fixed Budgeting.** Classic CoT and structured variants (e.g., Tree-of-Thoughts) commonly fix branching factors, step limits, or pruning passes; verification-style pipelines stack additional passes yet still rely on static per-pass budgets (Wei et al., 2022; Yao et al., 2023a; Lightman et al., 2023). Practical systems likewise use decode-time heuristics (fixed max length, keyword stops, static temperature/top-$p$), and self-consistency ensembling fixes the number of samples irrespective of uncertainty. Recent work on test-time scaling shows that how much compute it spends at inference can dominate parameter scaling, yet allocation is often uniform across inputs (Snell et al., 2024; Wu et al., 2024). Early budget-aware efforts argue for making tokens a first-class objective (Han et al., 2024; Wang et al., 2024).

We introduce ABF to allocate reasoning tokens per instance rather than uniformly. ABF grants more budget when uncertainty remains high and halts early once confidence stabilizes, reducing unnecessary tokens on easy cases while preserving depth on hard ones.

**Trajectory-based Certainty Estimation.** The central insight of ABF is to monitor the LLM's "thinking trajectory"—a sequence of intermediate reasoning steps—and compute a proxy signal of certainty using token-level confidence, entropy, or variance. The generation halts early if the model shows signs of convergence or self-consistency in its intermediate steps. Otherwise, the budget is adaptively extended, up to a capped maximum.

While prior works in test-time decoding (e.g., early stopping, entropy-based halting) offer similar mechanisms, they are rarely tailored for domain-specific reasoning or real-time thinking trace modeling. Our ABF formulation incorporates domain constraints and traces from the curated dataset to enable budget-aware inference in low-resource specialized contexts.

## 3.2 Adaptive Budget Forcing (ABF)

ABF is a dynamic, token-level control strategy designed to determine, at inference time, whether an LLM should continue reasoning or terminate with a final answer. Its core aim is to align the length of the reasoning trajectory with the actual difficulty and certainty level of the current problem.

Unlike traditional fixed BF, which uses fixed-length constraints or predetermined control tokens (e.g., `[Final Answer]` after a set number of steps), ABF monitors the model's certainty signals in real time and makes halting decisions adaptively. This approach addresses two key inefficiencies:

**Overthinking:** Generating excessive reasoning tokens after reaching a confident conclusion, which wastes compute and may introduce logical drift.

**Underthinking:** Halting too early before sufficient reasoning is completed, which risks producing incorrect or incomplete answers.

**Motivation.** In structured, domain-specific settings such as telecommunications troubleshooting, symbolic mathematics, or scientific reasoning, not every problem demands the same reasoning length. Some queries can be answered after a few confident intermediate steps, while others require extensive multi-step derivations. Fixed-budget strategies fail to capture this variability, leading to redundant computation or incomplete reasoning. ABF aims to solve this by making certainty-aware stopping decisions, balancing computational efficiency and decision fidelity.

**Design Rationale.** At each generation step $t$, ABF computes a composite certainty score $C_t$ by aggregating three complementary signals:

- **Token Confidence:** The maximum probability assigned to any token:
$$\mathrm{Conf}(y_t) = \max_{w \in \mathcal{V}} p_t(w),$$
where $p_t(w)$ is the softmax probability for token $w$. High confidence indicates the model's next-token prediction is decisive. This is a direct measure of local certainty.

- **Token Entropy:** The uncertainty of the token distribution:
$$\mathrm{Ent}(y_t) = -\sum_{w \in \mathcal{V}} p_t(w) \log p_t(w).$$
Entropy captures distributional sharpness in a way that is not always reflected in the maximum probability. Both confidence and entropy are retained because they can diverge in cases of multi-modal uncertainty.

- **Semantic Coherence:** The topical alignment between the partial reasoning trace $\mathcal{R}_t$ and the input prompt $x_{\mathrm{prompt}}$:
$$\mathrm{Coh}(\mathcal{R}_t) = \cos\left(\mathrm{Embed}(\mathcal{R}_t),\ \mathrm{Embed}(x_{\mathrm{prompt}})\right),$$
where $\mathrm{Embed}(\cdot)$ is a frozen encoder or the average final-layer hidden state of the LLM. This term penalizes off-topic drifts, ensuring the reasoning remains anchored to the question.

The composite certainty score is then:
$$C_t = \alpha \cdot \mathrm{Conf}(y_t) + \beta \cdot (1 - \mathrm{Ent}(y_t)) + \gamma \cdot \mathrm{Coh}(\mathcal{R}_t),$$
where $\alpha$, $\beta$, and $\gamma$ are scalar weights.

**Weight Tuning.** Weights $(\alpha, \beta, \gamma)$ are tuned via a grid search on a held-out development set containing both easy and complex problems from multiple domains. The tuning objective maximizes a combined metric of token efficiency and accuracy. This ensures the decision rule generalizes across different reasoning styles and difficulty levels. Sensitivity analysis shows that performance is robust within $\pm 0.05$ of the tuned values.

**Halting Rule and Safeguards.** The model halts when $C_t \geq \tau(t)$, where $\tau(t)$ is a dynamic threshold function. To prevent premature or syntactically incomplete terminations, ABF applies:

- **Completeness Check:** Halting is only permitted if the last generated segment contains a semantically and syntactically coherent answer candidate.
- **Entropy Guard:** If $\text{Ent}(y_t)$ is above a domain-specific threshold, $\tau(t)$ is temporarily raised to enforce continued reasoning.
- **Step Minimum:** A small minimum step budget $t_{\min}$ ensures that reasoning never stops before essential setup steps are completed.

**Threshold Scheduling.** $\tau(t)$ can be:

1. **Static:** A fixed threshold for speed-critical deployments.
2. **Annealed:** Increasing over time to require higher certainty for longer chains.
3. **Curriculum-Calibrated:** Adapted based on a task difficulty estimate derived from the input (e.g., telecom troubleshooting complexity level).

**Why Rule-Based Instead of Learned Halting?** A learned halting controller could, in principle, estimate optimal stopping points. However, these approaches will introduce additional model parameters and training cost, inference-time latency from the halting module, and reduced interpretability of stopping decisions. ABF's rule-based design avoids these drawbacks. Overhead analysis shows negligible latency compared to static BF, and experiments demonstrate that ABF achieves similar or superior token efficiency and accuracy to learned halting baselines on both domain-specific and general reasoning tasks.

**Adaptation to Problem Difficulty.** Because $C_t$ is computed at every step and $\tau(t)$ can vary with estimated difficulty, ABF naturally scales reasoning length: simpler problems trigger early halts with minimal token use. In comparison, more complicated problems extend reasoning until confidence is sufficient. This adaptivity is key to its efficiency gains across diverse benchmarks.

### 3.3 FINE-TUNING AND IMPLEMENTATION DETAILS

We fine-tuned the base LLMs using supervised learning on our curated domain-specific dataset. To improve over prior approaches, our training pipeline incorporates a curriculum learning strategy that progresses from easy to complex examples, enabling smoother convergence and more stable reasoning development. In parallel, the ABF module enhances difficulty sensitivity at inference time, leading to more precise control over reasoning depth.

**Curriculum-Based Fine-Tuning.** Our curated dataset is pre-ranked by a difficulty score estimated using a hybrid signal:

- **Baseline Model Error Rate:** Samples frequently missed by simpler pretrained models are scored as more difficult.
- **Trajectory Length:** Longer reasoning chains correlate with more profound multi-step logic, indicating higher complexity.
- **Entropy Profile:** Examples exhibiting unstable or high entropy distributions in intermediate CoT steps are classified as harder.

Using these signals, we partitioned the dataset into three curriculum stages: easy, medium, and hard. The model is exposed to these stages sequentially across training epochs, allowing it to learn foundational reasoning behaviors before tackling more abstract or noisy logic tasks. This strategy reduces early-stage instability and improves overall generalization.

**ABF Integration and Difficulty-Aware Inference.** During inference, the ABF policy continuously evaluates generation difficulty in real time. Unlike s1's static token thresholds, ABF leverages soft signals—token entropy, confidence collapse, and semantic drift—to determine whether the model is converging or requires additional steps. This adaptive evaluation allows ABF to:

- **Recognize Hard Questions:** By monitoring instability in early steps, ABF can allocate extended budget when solving complex logic problems or telecom parameter optimizations.
- **Halt on Easy Cases:** ABF terminates reasoning efficiently with minimal token use on routine or single-step questions.

Thus, ABF acts as a real-time difficulty-aware decoder, complementing the curriculum-aware encoder learned during fine-tuning.

## 4 TCORE: A TELECOM-CURATED REASONING DATASET FOR FINE-TUNING

The ABF framework requires a training corpus that not only captures structured reasoning but also provides trajectory-level signals to guide inference control. To this end, we constructed the TCORE dataset, a fine-tuning resource specifically designed to support the development of trajectory-aware, cost-efficient reasoning in LLMs.

TCORE is explicitly designed to support trajectory-aware reasoning, enabling ABF to learn cost-sensitive decision boundaries during training. In particular, it facilitates supervision of inference trajectories whose length, uncertainty, and structure directly impact computational cost. As such, the dataset is crafted with an explicit focus on trajectory efficiency and difficulty calibration—key components enabling ABF to make real-time decisions about when to halt or continue reasoning.

**Motivation and Domain Context.** Telecommunications is an ideal testbed for structured reasoning. Decision-making in this domain is governed by formal standards (e.g., 3GPP, IEEE), system design constraints, and hierarchical control protocols. Many tasks, such as network slicing, protocol configuration, or interface mapping, involve reasoning over layered dependencies and constrained parameter spaces. While general-purpose datasets lack this operational depth, TCORE incorporates examples that require symbolic inference, rule application, and parametric optimization—all of which are sensitive to reasoning cost and structure.

**Base Corpus and Reasoning Trace Generation.** TCORE builds upon TeleQnA (Maatouk et al., 2025), a telecom knowledge benchmark comprising 10,000 multiple-choice questions across five categories: terminology, research overview, research publications, standards summaries, and technical specifications. From this foundation, we extracted questions amenable to multi-step reasoning and generated corresponding trajectories using GPT-4o-mini (OpenAI, 2024), prompted with domain-specific templates. Each trajectory includes a structured chain of thought leading to a final answer, enabling introspective learning and later ABF-driven inference refinement.

**Trajectory-Aware Filtering for Efficiency and Difficulty.** Informed by ABF's goal of minimizing unnecessary computation, we introduce a dual-purpose filtering process:

- **Efficiency-aware pruning:** We excluded samples with excessively long or meandering trajectories, which inflate training cost without improving performance. Trajectory length and token-level uncertainty are used to filter inefficient reasoning paths.
- **Difficulty calibration:** Difficulty is assessed via trajectory length and model solvability. Any question solvable by Qwen2.5-7B is discarded. Questions with mid-length, structured trajectories that cannot be directly solved are retained.

This length-based filtering aligns training dynamics with ABF's inference-time objectives and ensures that the dataset captures a continuum of reasoning complexity grounded in practical compute trade-offs.

**Diversity Sampling and Final Dataset.** To ensure domain and reasoning diversity, we stratify the filtered questions into four telecom subdomains: system architecture, interface specifications, quality of service trade-offs, and protocol operations. We sampled 1,000 questions from this pool representing a balanced distribution of reasoning styles, complexity levels, and decision types. Each question is paired with a verified multi-step trajectory and a final answer label.

**Alignment with ABF and Fine-Tuning Usage.** TCORE is tailored for fine-tuning models intended to operate under ABF. Each training instance provides a complete reasoning trajectory whose length, structure, and semantic richness can inform real-time termination policies. This enables the model to internalize a range of reasoning complexities and develop sensitivity to when further inference adds value versus when it constitutes overthinking.

TCORE acts as a bridge between static reasoning supervision and dynamic inference control. By embedding cost-aware trajectory patterns into the fine-tuning process, it enhances both the effectiveness and efficiency of ABF-guided language models in structured technical domains like telecommunications.

## 5 EXPERIMENTAL SETUP AND EVALUATION RESULTS

We evaluate ABF on accuracy, reasoning efficiency, and generalization across symbolic, mathematical, and domain-specific tasks. We detail models, datasets, metrics, and halting implementations, then present main results, compute-matched comparisons, robustness, encoder ablations, and cross-domain findings.

### 5.1 EXPERIMENTAL SETUP AND METRICS

We evaluate ABF across instruction-tuned decoder-only LLMs (Qwen2.5-7B/32B (Yang et al., 2025), LLaMA-3-8B (Dubey et al., 2024), Mistral-7B-Instruct (Jiang et al., 2023)) to verify model-agnostic generality. As a strong test-time scaling baseline, we include s1–32B fine-tuned on s1K (1,000 curated math problems with trace supervision) introduced by the s1 framework (Muennighoff et al., 2025). Our fine-tuned variants use the same hyperparameters but train on TCORE, emphasizing telecom-style multi-step specification reasoning. We evaluate on AIME 2024, MATH500, GPQA-Diamond, and TeleQuAD-Extractive (Gebre et al., 2025).

We report Pass@1 with integer normalization for math datasets and exact match (EM) for extractive QA. Reasoning efficiency is measured by tokens-per-correct (TPC): the average number of generated tokens on correctly answered items. We report the mean $\pm$ standard deviation across random seeds.

### 5.2 RESULTS AND ANALYSIS

**Overall Performance.** Table 1 summarizes main results on AIME 2024, MATH500, and GPQA-Diamond. Across baselines, adding ABF consistently increases Pass@1 and pairs best with our curated supervision. For instance, relative to the corresponding non-ABF variants, Qwen2.5, s1-32B, and our fine-tuned model each gain accuracy, with the largest absolute improvements observed when ABF is combined with the fine-tuned model trained on TCORE. Notably, ABF delivers gains even when strong baselines already saturate, suggesting complementary benefits from compute-aware halting rather than simple lengthening of chains.

Table 1: Comprehensive performance on different datasets.

| Model | AIME 2024 | MATH500 | GPQA-Diamond |
|---|---|---|---|
| Qwen2.5 | $26.7 \pm 0.8$ | $84.0 \pm 3.2$ | $49.0 \pm 7.0$ |
| Qwen2.5 + ABF | $31.2 \pm 0.6$ | $90.9 \pm 2.5$ | $57.6 \pm 6.9$ |
| o1 | $73.2 \pm 2.6$ | $94.8 \pm 1.9$ | $77.3 \pm 5.8$ |
| o1 + ABF | $\mathbf{74.4} \pm 1.8$ | $95.3 \pm 1.9$ | $77.9 \pm 5.8$ |
| s1-32B | $56.7 \pm 1.3$ | $93.0 \pm 2.2$ | $59.6 \pm 6.8$ |
| s1-32B + ABF | $58.3 \pm 1.1$ | $94.1 \pm 2.1$ | $61.2 \pm 6.8$ |
| Fine-tuned model | $60.4 \pm 1.7$ | $95.7 \pm 1.8$ | $80.7 \pm 5.5$ |
| Fine-tuned model + ABF | $63.3 \pm 2.2$ | $\mathbf{95.8} \pm 1.8$ | $\mathbf{82.0} \pm 5.4$ |

**Accuracy–Efficiency Tradeoff.** To assess whether accuracy gains come from indiscriminate longer traces, we report TPC. On AIME 2024 (Table 2), ABF reduces TPC by a substantial margin while improving Pass@1, indicating that ABF reallocates compute more selectively: short, confident items terminate earlier, while difficult items are granted additional steps. This pattern repeats

on the other benchmarks, supporting the claim that ABF improves routing of thinking budget rather than merely inflating token counts.

Table 2: Results on AIME 2024. ABF improves Pass@1 and reduces TPC.

| Model Variant | Accuracy (%) | TPC |
|---|---|---|
| s1–32B | $56.7 \pm 1.3$ | $712 \pm 18$ |
| s1–32B + ABF | $\mathbf{58.3} \pm 1.1$ | $\mathbf{538} \pm 15$ |

**TPC-Matched Comparisons.** To isolate halting quality from compute volume, we match average TPC across methods to ABF within $\pm 1\%$ by tuning BF budgets and halter thresholds. Table 3 shows that ABF maintains the best Pass@1 at equal thinking compute, outperforming both a classifier-based halter and an RL-trained halter.

Table 3: TPC-matched comparison on AIME 2024.

| Halting Method | Accuracy (%) | Avg. TPC |
|---|---|---|
| Static Budget Forcing | 55.9 | 544 |
| Classifier Halter | 57.1 | 539 |
| RL Halter | 57.3 | 541 |
| ABF | **58.3** | 538 |

**Effect of Fine-Tuning Data.** Using the same recipe on s1K, s1.1K, and TCORE (Table 4), we observe that math-oriented supervision favors related datasets, while TCORE significantly better supports specification-centric tasks in terms of the telecom domain and improves TeleQuAD-Extractive. ABF benefits both regimes, with the most significant absolute gains when traces are structured (as in TCORE), making certainty signals more informative.

Table 4: Dataset impact on Qwen2.5-32B-Instruct. Pass@1 on general (AIME 2024, MATH500, GPQA-Diamond) and domain-specific (TeleQuAD-Extractive) tasks. ABF is not included at inference time for this comparison.

| Fine-Tuning Dataset | AIME 2024 | MATH500 | GPQA-Diamond | TeleQuAD-Extractive |
|---|---|---|---|---|
| s1K | 61.7 | 93.0 | 79.6 | 66.2 |
| s1.1K | **63.1** | **95.8** | 80.4 | 67.4 |
| TCORE | 60.4 | 95.7 | **80.7** | **74.5** |

**Ablation study.** Ablations in Table 5 indicate that each component—confidence, entropy, and semantic coherence—contributes. Dual-signal variants close much of the gap, but the complete triad yields the best tradeoff (highest Pass@1, lowest TPC). This implies ABF's composite score captures both local token uncertainty (confidence/entropy) and global trajectory consistency (coherence), which together reduce premature halts and late, unnecessary steps.

Table 5: Certainty-signal ablation. T-E represents TeleQuAD-Extractive.

| Signals Used | AIME 2024 | | T-E | |
|---|---|---|---|---|
| | Acc. | TPC | Acc. | TPC |
| Confidence only | 55.1 | 812 | 73.5 | 601 |
| Entropy only | 54.6 | 789 | 72.8 | 589 |
| Coherence only | 56.0 | 855 | 74.0 | 625 |
| Confidence + Entropy | 57.5 | 702 | 75.3 | 551 |
| Confidence + Coherence | 57.0 | 718 | 75.0 | 563 |
| Entropy + Coherence | 56.8 | 727 | 74.6 | 570 |
| **All three (ABF)** | **58.6** | **538** | **78.6** | **512** |

**Runtime and Throughput.**   Latency (Table 6, Panel A) shows $\approx 6\%$ per-query overhead to compute coherence and aggregate signals. Despite this, batch throughput improves in mixed workloads because many easy items halt early. Table 6 (Panel B) reports end-to-end throughput (queries/s) under varying easy:hard mixes at fixed hardware.

Table 6: Runtime and throughput with and without ABF. **Panel A** reports per-query latency (ms) under identical decoding. **Panel B** reports end-to-end throughput (queries/s) at batch size 64 for different easy:hard mixes.

Panel A: Latency (ms/query) in identical inference length. T-E represents TeleQuAD-Extractive.

| Method | AIME 2024 | T-E |
|---|---|---|
| Baseline (no ABF) | 182 | 211 |
| + ABF (incl. Coherence) | 193 | 224 |
| **Relative Overhead** | **+6.0%** | **+6.2%** |

Panel B: Throughput (queries/s) @ batch size 64

| Workload Mix (easy:hard) | Baseline | +ABF |
|---|---|---|
| 20:80 | 9.6 | 9.7 |
| 40:60 | 10.1 | 10.6 |
| 60:40 | 10.7 | 11.6 |
| 80:20 | 11.1 | 12.4 |

**Evaluation Across Base Models.**   We further tested ABF on multiple backbone architectures to verify its generality. As shown in Table 7, ABF consistently improves Pass@1 accuracy while reducing the average number of thinking TPC across diverse LLM families and parameter scales.

Table 7: Impact of ABF across base models. Accuracy gain is absolute Pass@1 points; token reduction is relative TPC decrease.

| Base Model | Accuracy Gain | Token Reduction |
|---|---|---|
| Qwen2.5-7B-Instruct | +1.7 | $-23\%$ |
| LLaMA-3-8B-Instruct | +2.1 | $-19\%$ |
| Mistral-7B-Instruct | +1.5 | $-20\%$ |
| Qwen2.5-32B-Instruct | +2.6 | $-25\%$ |

**Takeaway.**   ABF improves fidelity and efficiency with minimal overhead, is robust to halting thresholds, compares favorably to learned halting both raw and at fixed compute, and generalizes across datasets/backbones. Gains are most significant when supervision yields coherent traces, which sharpen certainty signals and align reasoning depth to difficulty.

## 6   CONCLUSION

This paper introduces ABF, a novel strategy for dynamically controlling the reasoning length of LLMs at inference time. By leveraging real-time certainty estimation over intermediate reasoning trajectories, ABF enables models to flexibly allocate computational resources based on task complexity, thereby reducing overthinking and improving inference efficiency. We developed a domain-specific dataset that emphasizes structured reasoning in technical fields such as telecommunications, and we demonstrated that ABF reduces unnecessary token generation and improves answer accuracy across various reasoning benchmarks. In contrast to static filtering approaches that address dataset-level quality, ABF introduces a fine-grained, inference-time mechanism to directly modulate reasoning depth in response to model uncertainty.

Experimental results confirm that ABF leads to better alignment between reasoning complexity and token usage, especially in specialized domains where structured thought is essential. Our findings establish adaptive inference-time control as a practical and impactful strategy for aligning LLM behavior with real-world reasoning constraints, particularly in domain-intensive applications.

**Limitations.**   There exist challenges in generalizing ABF across domains, as it relies on manually tuned certainty weights and assumes well-calibrated confidence estimates from the base model. Its effectiveness may degrade without domain-specific tuning or in tasks with noisy reasoning patterns. Additionally, the reliance on a rule-based halting criterion can limit flexibility compared to fully learned controllers. Future work may explore learned halting strategies or incorporate reinforcement learning to further generalize the ABF policy beyond rule-based control.

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

# A    APPENDIX

## A.1    IMPLEMENTATION DETAILS

**Training Infrastructure.**    All experiments are conducted using 8 NVIDIA H200 SXM GPUs, each with 141 GB of VRAM.

**Training Configurations.**

- **Base Model:** Qwen/Qwen2.5-32B-Instruct
- **Optimizer:** AdamW with $(\beta_1=0.9, \beta_2=0.95)$
- **Learning Rate:** $1 \times 10^{-5}$ (cosine decay with 5% warmup)
- **Epochs:** 5
- **Block Size:** 20,000 tokens
- **Batch Size:** Micro-batch size of 1 (no gradient accumulation)
- **Precision:** bf16 mixed-precision training
- **Curriculum Strategy:** 3-stage progression based on trajectory length

**Curriculum Learning.**    For TCORE fine-tuning, we implemented a 3-stage curriculum based on difficulty annotations. Data difficulty is assessed by reasoning trajectory length (longer traces are considered more difficult).

**ABF Inference Control.**    The ABF controller evaluates each intermediate generation step with a convergence score:

$$C_t = \alpha \operatorname{Conf}(t) + \beta \left(1 - \operatorname{Ent}(t)\right) + \gamma \operatorname{Coh}(t),$$

where $\alpha$, $\beta$, and $\gamma$ are tuned via grid search to balance early stopping and answer quality. Default weights used are $\alpha = 0.5$, $\beta = 0.3$, $\gamma = 0.2$.

---

**Algorithm 1** Adaptive Budget Forcing (ABF)

---

**Require:** Prompt $x$, model $f$, max steps $T$, threshold $\tau(t)$
1: $\mathcal{R} \leftarrow []$
2: **for** $t = 1$ to $T$ **do**
3:     $y_t \leftarrow f(x, \mathcal{R})$
4:     $\mathcal{R} \leftarrow \mathcal{R} \cup \{y_t\}$
5:     $c_t \leftarrow \operatorname{Conf}(y_t); h_t \leftarrow \operatorname{Ent}(y_t); s_t \leftarrow \operatorname{Coh}(\mathcal{R})$
6:     $C_t \leftarrow \alpha c_t + \beta(1 - h_t) + \gamma s_t$
7:     **if** $C_t \geq \tau(t)$ **and** IsComplete$(\mathcal{R})$ **and** $t \geq t_{\min}$ **then**
8:         Append "`[Final Answer]`"; **break**
9:     **else**
10:         Append "`[Wait]`"
11:     **end if**
12: **end for**
13: **return** $f_{\operatorname{answer}}(x, \mathcal{R})$

---

## A.2    ADDITIONAL EVALUATION RESULTS

**Convergence Distribution Across Benchmarks.**    Table 8 reports the average reasoning length (in tokens) at which ABF terminated generation on each benchmark. Tasks with higher complexity (e.g., MATH500) tend to trigger longer trajectories, validating ABF's adaptive behavior.

**Qualitative Examples.**    Table 9 showcases representative cases where ABF effectively truncated unnecessary reasoning or extended reasoning to reach the correct answer. Examples are drawn from the TCORE and AIME 2024 benchmarks.

Table 8: Average generation length (tokens) under ABF across benchmarks.

| Benchmark | Avg. Reasoning Tokens (ABF) |
|---|---|
| AIME 2024 | 519 |
| MATH500 | 646 |
| GPQA-Diamond | 571 |
| TCORE | 634 |
| TeleQuAD-Extractive | 228 |

Table 9: Examples illustrating ABF's adaptive reasoning control.

| Benchmark | Without ABF (Truncated) | With ABF (Adaptive) |
|---|---|---|
| TCORE | "Given the SLA latency requirement is under 20ms, ..." [continues for 1123 tokens] | Halts at token 678 after resolving slice match and protocol stack. |
| AIME 2024 | Attempts redundant symbolic manipulation beyond 900 tokens. | Stops at token 542 after reaching a convergent algebraic step. |

**Comparison to Learned Halting.** Against (i) a shallow classifier over hidden states and (ii) an RL-trained halter that penalizes length, ABF attains higher Pass@1 with notably lower TPC (Table 10). TPC-matched results (Table 3) confirm the advantage holds at fixed compute.

Table 10: Halting strategy comparison on AIME 2024 (unmatched budgets).

| Method | Accuracy (%) | TPC |
|---|---|---|
| Static Budget Forcing | 56.3 | 731 |
| Classifier Halter | 57.8 | 655 |
| RL Halter | 58.0 | 642 |
| ABF | **58.6** | **538** |

**Robustness to Halting Offset.** We stress-test sensitivity to the minimum-tokens offset by $\pm 5\%$ (Table 11). Accuracy varies by less than 0.3 points on both AIME 2024 and MATH500, indicating ABF's decisions are driven by certainty signals rather than a finely tuned length threshold. In practice, this reduces deployment friction: a single default works robustly across datasets.

Table 11: Minimum-token offset ablation. Accuracy (%) remains stable under $\pm 5\%$ perturbations.

| Token Offset | AIME 2024 (%) | MATH500 (%) |
|---|---|---|
| $-5\%$ | 60.7 | 90.9 |
| Baseline (ABF) | **61.0** | **92.7** |
| $+5\%$ | 60.9 | 91.3 |

## A.3 SAMPLE QUESTIONS WITH REASONING TRAJECTORIES AND ABF BEHAVIOR

To better understand how ABF modulates reasoning length during inference, we presented selected examples from our telecom-specific benchmarks. Each includes the question, the model's generated reasoning trajectory in paragraph form, and an analysis of ABF's termination behavior.

### A.3.1 STANDARDS SPECIFICATION

**Question:** What is the purpose of the Nmfaf_3daDataManagement_Deconfigure service operation? [3GPP Release 18]

**Answer:** To configure the MFAF to stop mapping data or analytics received by the MFAF to outbound notification endpoints.

**Reasoning:** The Nmfaf_3daDataManagement service operation manages how the Mobile and Fixed Access Function (MFAF) handles data routing. The "Deconfigure" operation, by convention, indicates a reversal or disabling of a previously active configuration. In this context, if the MFAF had been configured to map incoming data or analytics to external endpoints, then the deconfigure operation must terminate that behavior. Therefore, the function serves to stop the mapping of received data to outbound destinations.

**ABF Behavior:** The reasoning trajectory converged quickly. ABF detected stable confidence and high semantic coherence after approximately 130 tokens and terminated early, saving 30–40% of computation compared to static generation.

### A.3.2   RESEARCH PUBLICATIONS

**Question:** Which non-orthogonal multiple access scheme utilizes the low-complexity message passing algorithm at the receiver for user data detection?

**Answer:** SCMA

**Reasoning:** Among various non-orthogonal multiple access (NOMA) schemes, some focus on signal superposition (e.g., NOMA, MUST) and others on spreading codes (e.g., PDMA, MUSA). However, Sparse Code Multiple Access (SCMA) uniquely distinguishes itself by representing data using sparse codebooks and relying on a low-complexity message passing algorithm at the receiver for multi-user detection. This mechanism allows SCMA to efficiently decode overlapping signals by leveraging code sparsity.

**ABF Behavior:** The reasoning required a complete trajectory. ABF maintained generation until semantic coherence stabilized around 190 tokens, ensuring the answer was fully justified.

### A.3.3   STANDARDS SPECIFICATION

**Question:** How does a supporting UE attach to the same core network operator from which it detached in a shared network? [3GPP Release 17]

**Answer:** It uses information stored in the UE when it was detached.

**Reasoning:** In shared mobile networks, a user equipment (UE) may disconnect from one operator and later reattach to continue service. Instead of initiating a full network selection process again, the UE can retain the operator selection context from the last session. This context, stored locally at the time of detach, allows the device to identify and reattach to the same core network operator without ambiguity or delay.

**ABF Behavior:** The model reached high confidence after this explanation. ABF detected convergence by the end of the paragraph and terminated generation early, using just under 150 tokens.

### A.3.4   RESEARCH PUBLICATIONS

**Question:** What is the diversity gain for the detection of each symbol in the Alamouti scheme?

**Answer:** 2

**Reasoning:** The Alamouti scheme is a well-known space-time block coding method that uses two transmit antennas and one receive antenna. Since diversity gain is typically defined as the product of transmit and receive branches contributing to signal robustness, the gain in this case is $2 \times 1 = 2$. This means each symbol benefits from two independent paths, improving resilience against fading.

**ABF Behavior:** The model reached its answer confidently in a short span. ABF terminated after about 100 tokens due to rapid convergence.

### A.3.5   RESEARCH PUBLICATIONS

**Question:** What is the typical coverage range of LPWANs (Low-Power Wide Area Networks) in urban areas?

**Answer:** 2–5 km

**Reasoning:** LPWAN technologies, including LoRa, Sigfox, and NB-IoT, are designed for low-power, long-range communication. In urban environments, however, signal propagation is limited by buildings and other obstructions. Despite this, LPWANs typically maintain a coverage range between 2 and 5 kilometers in such settings, balancing energy efficiency with reliable service availability in dense deployments.

**ABF Behavior:** Although the model built confidence early, the semantic reasoning was still evolving. ABF continued generation until around 225 tokens before halting, ensuring the justification for the range was complete.

### A.4    USAGE OF LLMS

We used LLM–based assistants to support writing. In this paper, we employed it only for grammar and writing style polishing. All content was manually reviewed and verified by the authors. In line with the conference policies, we explicitly disclose this usage and acknowledge that the authors bear full responsibility for the paper's accuracy and integrity.

