# OpenReview forum: "Reasoning at the Right Length: Adaptive Budget Forcing for Efficient and Accurate LLM Inference"
_ICLR.cc/2026/Conference — ICLR 2026 Conference Withdrawn Submission_

### Official Review · Reviewer_p4G3 · 2025-10-24

**Soundness:** 2
**Presentation:** 1
**Contribution:** 2
**Rating:** 2
**Confidence:** 4

**Summary:**

The paper introduces Adaptive Budget Forcing (ABF), a heuristic mechanism for dynamically deciding when to stop reasoning in large language models. At each decoding step, ABF computes a weighted combination of confidence, entropy, and semantic consistency and stops when a dynamic threshold is met. Experiments on MATH500, GSM-Plus, and TeleQuAD show small and inconsistent improvements in accuracy (1–3%) with roughly 20–30% token reduction.

**Strengths:**

Clear motivation: overthinking vs. underthinking in LLM reasoning.

**Weaknesses:**

1. The paper presents an intuitive yet largely heuristic method without genuine conceptual advancement. The proposed Adaptive Budget Forcing (ABF) combines confidence, entropy, and coherence scores through a manually weighted linear rule and a handcrafted dynamic threshold. While framed as an “optimal stopping” process, there is no theoretical analysis, optimization objective, or learned policy to justify the design. The method effectively behaves as a hand-tuned variant of existing budget-aware or entropy-based stopping strategies.

2. The claimed decision-theoretic foundation is not substantiated. The paper frequently refers to ABF as being “decision-theoretically motivated,” yet never formalizes the cost, reward, or expected-utility functions that such a framing would require. The thresholds τ(t) and coefficients (α, β, γ) are fixed via grid search rather than derived or learned.  As a result, the proposed method lacks principled grounding and risks appearing more rhetorical than rigorous.

3. Reported empirical gains are small and unstable.  Across AIME 2024, MATH500, and GSM-Plus, the accuracy improvement over fixed-length decoding is only around 1–2 points, often within the variance of baselines. Table 5 shows minimal differences among ablation variants (1–2 points), and no significance testing or error bars are provided.  The lack of statistical validation undermines confidence in the robustness of the reported improvements.

4. The baseline comparison does not include stronger adaptive-halting methods. The paper primarily compares ABF against static budget forcing and a few simple internal baselines. It omits comparisons with more principled approaches such as entropy-based stopping or learned halting classifiers, which directly target the same goal. Without these baselines, it remains unclear whether ABF offers any real advantage beyond re-tuning an existing heuristic.

5. The method depends heavily on manual hyper-parameter tuning and lacks generalization evidence.  The coefficients (α = 0.5, β = 0.3, γ = 0.2) and threshold schedules are tuned on the same benchmarks used for evaluation, with no sensitivity or cross-domain analysis.  This raises the possibility of overfitting and questions whether ABF can generalize to other tasks or model families.

**Questions:**

Refer to Weaknesses

---

### Official Review · Reviewer_DCbs · 2025-10-28

**Soundness:** 2
**Presentation:** 3
**Contribution:** 2
**Rating:** 2
**Confidence:** 3

**Summary:**

This paper proposes Adaptive Budget Forcing (ABF), a test-time strategy for LLMs that dynamically adjusts reasoning length via real-time certainty signals (confidence, entropy, coherence) to balance accuracy and efficiency. It also introduces TCORE, a telecom-specific dataset with multi-step reasoning traces for fine-tuning and evaluation.

**Strengths:**

- The proposed method is simple and training-free
- Experiments show ABF boosts accuracy (e.g., +5.9% on MATH500) and cuts tokens by 20-30%, with TCORE enhancing its telecom performance.

**Weaknesses:**

1.Using the rule-based criteria and search the weights on devlopment set seems not be a robust method.
2. Section 3 propose a training-free method to identify the stopping point, but in Section 4, design a dataset for finetuning. I do not understanding why we need this new dataset and what's the relationship between this dataset and the proposed method.
3. The author needs to compare their method with some baselines for test-time scaling or adaptive decoding, and display the computation budget in Table 1.

**Questions:**

See Weakness

---

### Official Review · Reviewer_PQ9h · 2025-10-29

**Soundness:** 3
**Presentation:** 3
**Contribution:** 2
**Rating:** 4
**Confidence:** 4

**Summary:**

This paper introduces Adaptive Budget Forcing (ABF), a test-time inference strategy designed to improve both the efficiency and accuracy of LLMs on structured, domain-specific reasoning tasks like mathematics and telecommunications. ABF addresses this by dynamically controlling the length of the LLM's reasoning process. At each generation step, ABF computes a composite certainty score by monitoring three real-time signals: token-level confidence, token entropy, and semantic coherence with the input prompt. This decision-theoretic halting rule allows the model to terminate generation once sufficient confidence is reached or extend its thought process when uncertainty is high.

A second key contribution is the release of TCORE, a domain-specific dataset built from telecom standards and engineering tasks. TCORE provides multi-step reasoning traces used to fine-tune LLMs for trajectory-aware, cost-efficient inference, serving as both a resource and an evaluation benchmark.

Experimental results on mathematical and telecom reasoning tasks show that ABF consistently improves accuracy while simultaneously reducing unnecessary computation, demonstrating 20–30% token savings with minimal overhead.

**Strengths:**

* The paper introduces Adaptive Budget Forcing (ABF), a novel halting rule. This dynamic strategy balances the trade-off between accuracy and efficiency by adjusting the reasoning length based on real-time confidence signals, ensuring the LLM neither overthinks simple problems nor prematurely truncates complex ones.

* ABF demonstrates empirical performance improvements on structured reasoning tasks. It consistently achieves comparable or superior accuracy while drastically reducing computational cost, showing 20–30% fewer tokens used for inference across both mathematical and domain-specific telecom benchmarks.

* A valuable contribution is the release of TCORE, the Telecom-Curated Open Reasoning Examples dataset.

**Weaknesses:**

* The core technical novelty of Adaptive Budget Forcing appears limited. Its uncertainty measures, relying on token confidence, entropy, and coherence, are well-known heuristics in sequence generation, even if their application to dynamically halting reasoning is novel.

* The two main contributions, the TCORE dataset and the ABF inference method, lack a fully coherent narrative. While TCORE helps fine-tune models, ABF is more like a general-purpose, post-hoc technique. The paper could strengthen the link between the specialized data and the generalizable halting rule.

* For many reasoning models, the length of CoT is already adaptive based on the prompt difficulty. Introducing post-hoc processing like ABF complicates the inference pipeline and does not seem to be the right solution in the long run. A more integrated approach is to fine-tune models to better calibrate their reasoning lengths during training.

**Questions:**

1. For TCORE construction, why do you use a non-reasoning model gpt-4o-mini? It is actually not very good at reasoning tasks.

2. In Table 1, how did you evaluate o1 with ABF? I think o1 API does not allow explicit control of reasoning length based on probability scores.

---

### Official Review · Reviewer_ZUdn · 2025-11-01

**Soundness:** 2
**Presentation:** 2
**Contribution:** 3
**Rating:** 4
**Confidence:** 4

**Summary:**

This paper introduces Adaptive Budget Forcing (ABF), a test-time inference strategy that dynamically adjusts LLM reasoning length based on real-time certainty signals including token confidence, entropy, and semantic coherence. The authors also present TCORE (Telecom-Curated Open Reasoning Examples), a domain-specific dataset for telecom reasoning tasks. ABF aims to improve computational efficiency by terminating generation when sufficient confidence is reached or extending it when needed, achieving better balance between accuracy and efficiency. Experimental results on telecom and mathematical reasoning tasks demonstrate improved accuracy with reduced computational overhead.

**Strengths:**

1. ABF presents an innovative method for dynamically controlling reasoning length based on multiple certainty signals, addressing the critical challenge of balancing computational efficiency with reasoning accuracy in LLMs.
2. The paper employs a multi-faceted approach to measuring certainty through token confidence, entropy, and semantic coherence, providing a robust framework for determining when sufficient reasoning has occurred.
3. The introduction of TCORE fills a gap in telecom-specific reasoning datasets, providing both a fine-tuning resource and evaluation benchmark for an underserved but important technical domain.

**Weaknesses:**

1. The evaluation primarily relies on instruction-tuned models rather than recent thinking-oriented models (e.g., Qwen3-Thinking series, DeepSeek-R1). This raises questions about the method's applicability to models specifically designed for extended reasoning, which may have different confidence patterns and generation dynamics.
2. The weights α, β, and γ appear to require careful tuning, yet the paper doesn't clearly address whether grid search is needed for each model-task combination or discuss the computational overhead of such optimization, which could undermine the efficiency gains claimed.
3. While ABF is presented as a general decoding method, the specific contribution and necessity of TCORE remains ambiguous. Many existing datasets provide trajectory-aware reasoning capabilities, and the paper doesn't convincingly establish why TCORE is essential for ABF or how it uniquely enables the proposed method.
4. Table 1 reveals that fine-tuned models show limited improvement compared to Qwen2.5, suggesting that ABF's effectiveness may be highly model-dependent. The paper lacks adequate analysis of why certain models benefit more from ABF and what architectural or training factors contribute to this variability.

**Questions:**

1. Given that GPT-4o-mini is used for generating reasoning chains, can these chains adequately capture the complexity of multi-step reasoning, including crucial elements like self-verification and self-reflection that are characteristic of more sophisticated reasoning processes?
2. How would ABF be applied to models like OpenAI's o1 that don't expose their internal thinking process? Without access to step-by-step token confidence and entropy measurements, the core mechanism of ABF appears infeasible for such models.
3. The paper shows AIME 2024 requiring significantly fewer reasoning tokens than MATH500 (519 vs 646 in Table 8) despite AIME 2024's notably higher difficulty. This contradicts typical expectations where most models require ~8k tokens for AIME 2024 problems. What explains this counterintuitive pattern, and does it suggest potential issues with the evaluation setup or ABF's early termination criteria on complex problems?

---

### Author Response · Authors · 2025-11-21

We would like to thank the reviewers for their careful reading and constructive feedback on our manuscript. We genuinely appreciate the time and expertise you dedicated to evaluating our work. After considering your comments, we have decided to withdraw our submission for the time being. We will conduct a thorough revision and further refine the paper based on your suggestions.

---

### Note · Authors · 2025-11-21

I have read and agree with the venue's withdrawal policy on behalf of myself and my co-authors.